# The First Report and Phylogenetic Analysis of Canine Distemper Virus in *Cerdocyon thous* from Colombia

**DOI:** 10.3390/v14091947

**Published:** 2022-09-01

**Authors:** Diego Fernando Echeverry-Bonilla, Edwin Fernando Buriticá-Gaviria, Delio Orjuela-Acosta, Danny Jaír Chinchilla-Cardenas, Julian Ruiz-Saenz

**Affiliations:** 1Hospital Veterinario, Universidad del Tolima, Calle 20 Sur # 23A-160 Barrio Miramar, Ibagué 730010, Tolima, Colombia; 2Grupo de Investigación en Medicina y Cirugía de Pequeños Animales, Facultad de Medicina Veterinaria y Zootecnia, Universidad del Tolima, Calle 20 Sur # 23A-160 Barrio Miramar, Ibagué 730010, Tolima, Colombia; 3Mascolab, Laboratorio de Biología Molecular, Calle 49 Sur # 45ª-300, Oficina 1202, Centro Empresarial S48 Tower, Envigado 055422, Antioquia, Colombia; 4Grupo de Investigación en Ciencias Animales—GRICA, Facultad de Medicina Veterinaria y Zootecnia, Universidad Cooperativa de Colombia, Bucaramanga 680002, Colombia

**Keywords:** canine distemper virus, interspecies transmission, Crab-eating fox, phylogeny, morbillivirus

## Abstract

**Simple Summary:**

Canine distemper virus (CDV) is the etiological agent of a highly frequent viral disease of domestic and wild carnivores. It poses a threat for the conservation of endangered species. Our aim was to assess the presence and phylogenetic characterization of CDV from naturally infected Crab-eating Fox (*Cerdocyon thous*) from Colombia. We confirm for the first time the circulation of CDV South America/North America-4 Lineage in Crab-eating Fox. Our results are crucial for the understanding of the interspecies transmission of CDV in the domestic/wild interface and for the prevention and control of such an important multi-host pathogen.

**Abstract:**

Canine distemper virus (CDV) is the etiological agent of a highly prevalent viral infectious disease of domestic and wild carnivores. This virus poses a conservation threat to endangered species worldwide due to its ability to jump between multiple species and produce a disease, which is most often fatal. Although CDV infection has been regularly diagnosed in Colombian wildlife, to date the molecular identity of circulating CDV lineages is currently unknown. Our aim was to evaluate the presence and phylogenetic characterization of CDV detected in samples from naturally infected *Cerdocyon thous* from Colombia. We sequenced for the first time the CDV infecting wildlife in Colombia and demonstrated the presence of South America/North America-4 Lineage with a higher relationship to sequences previously reported from domestic and wild fauna belonging to the United States of America. Our results are crucial for the understanding of the interspecies transmission of CDV in the domestic/wild interface and for the prevention and control of such an important multi-host pathogen.

## 1. Introduction

Canine distemper virus (CDV) is the etiological agent of a highly prevalent viral infectious disease of domestic and wild carnivores [1,2]. It is a negative-sense RNA virus belonging to the *Paramyxoviridae* family and Morbillivirus genus, which contains viruses with epidemiological relevance to humans and animals [3,4]. Based on H-gene sequences, CDV strains have been classified into at least 21 major genetic lineages: America-1, America-2, North America-3, South America/North America-4, America-5, Canada 1 and 2, Asia-1, Asia-2, Asia-3, Asia-4, Asia-5, Asia 6, Europe Wildlife, Arctic, Africa-1, Africa-2, Europe-1/South America-1, South America-2, South America-3 and Rockborn-like [5,6,7,8,9,10,11].

CDV creates a conservation threat to endangered species globally due to its ability to jump between multiple species [1,12]. To date, the virus has been found in multiple families and species of animals including the endangered Siberian tiger (*Panthera tigris altaica*) and the vulnerable giant panda (*Ailuropoda melanoleuca*) [13]. As a multi-host pathogen, CDV can emerge and re-emerge at the wildlife-domestic animal interface. The presence of CDV in wildlife has been well-established even in the absence of simultaneous infection in domestic populations, mainly driven by the existence of specialist vs. generalist strains [14]. CDV outbreaks in lions were shown to be asynchronous with viral circulation in domestic dogs, suggesting that CDV may persist in wild species interacting with multiple populations [15]. The critical role of wild animal reservoirs and meta-reservoirs is still unclear for most of the interspecies transmission scenarios of CDV [14]. However, the role of wildlife in intercontinental transmission of CDV lineage South America/North America-4 has been hypothesized due to its presentation in wild animals in the USA and in domestic dogs in Colombia, Peru and Ecuador [12].

Although CDV infection has been regularly diagnosed in Colombian wildlife, to date the molecular identity of circulating CDV lineages is currently unknown. Our aim was to evaluate the clinical features associated with canine distemper in Crab-eating foxes (*Cerdocyon thous*) in Colombia and to compare molecular and phylogenetic relationships between the virus sequences from these foxes with previously reported CDV sequences from domestic dogs from Colombia and the Americas.

## 2. Materials and Methods

### 2.1. Clinical Samples

In December 2021, three adult Crab-eating foxes (*Cerdocyon thous*) were found sick and reported to the Regional Ambiental Authority (CORTOLIMA) for assessment and medical care. Each fox was found in a different municipality of the Tolima Department (Figure 1) located in a Central Andean Region of Colombia (South America) (altitude range 314–807 m.a.s.l). A complete physical examination of each fox was performed by the attending veterinarian at the Veterinary Teaching Hospital (VTH) of the Tolima University.

### 2.2. Sample Collection

Blood samples were collected through jugular puncture using 5 mL syringes attached to a 25 G × 6 mm needle after disinfection of the jugular area. A total volume of 3–5 mL of blood was obtained from each fox and divided in 3 aliquots. One aliquot was added in labeled sample tube containing EDTA (Impromini^®®^, Guangzhou, China) as an anticoagulant and processed for a routine Complete Blood Count (CBC), performed in an automatic hematology analyzer (Fuerte Care FC-620^®®^, Palmira, Colombia). A second aliquot was added in sample tube with clot activator (BD Vacutainer^®®^ SST™, BD Life Sciences, NJ, USA), serum was collected and processed for serum biochemical analyses panel (creatinine, AST, ALT, and ALP), by using an automated analyzer (Biosystems A15 ^®®^, Barcelona, España). The third aliquot was added in labeled sample tube containing EDTA as anticoagulant and processed for CDV serological and rt-PCR tests. In all cases, the samples were identified and sent to the laboratory stored in climatized polystyrene boxes containing recyclable ice (4–8 °C).

### 2.3. Serological Rapid Test for CDV

A biochemical immunochromatography rapid test for CDV (CDV Ag Test Kit^®®^, Hwaseong-si, Korea) was performed following the company’s instructions. Briefly, the plasma sample and the rapid test kit were left at room temperature for 15 min prior to testing. After this time, one drop of serum was added to the sample buffer and mixed uniformly. With a pipette included in the kit, five drops of the prepared sample were applied to the sample slot of the cassette. Results were interpreted in 10–15 min.

### 2.4. RNA Extraction and cDNA Synthesis

Viral RNA was extracted from 140 µL of the serum by using QIAamp Viral RNA Mini Kit (QIAGEN^®®^, Hilden, Germany) following the manufacturer’s recommendations. RNA quality and quantity was determined by using NanoDrop™ One (Thermo Scientific, Delaware, MD, USA). Quantified RNA was preserved at −80 °C. The RevertAid™ First Strand cDNA Synthesis Kit (Thermo Scientific^®®^, Glen Burnie, MD, USA) were employed for cDNA synthesis. In brief, the mixture included 1 µL dNTP Mixture (10 mM), 1 µL (100 pmol/µL) random hexamers, and 13 µL (1–3 µg) total RNA. The mixture was heated for 5 min at 65 °C and next placed on ice. RT mixture contained of 4 µL Buffer 5X Reverse Transcriptase and 1 µL RevertAid™ Premium Enzyme Mixture. Reverse transcription was completed for 10 min at 25 °C, 30 min at 50 °C and 85 °C for 5 min.

### 2.5. PCR and Sequencing

Samples from foxes were tested by using “Universal” morbillivirus PCR for the phosphoprotein (P) gene [16]. The Maxima Hot Start Green PCR Master Mixture (Thermo Scientific^®®^) were used. In brief, 5 µL cDNA was included in a reaction mixture, which contained of 25 µL Master Mixture (2X), 15 µL nuclease-free water, and 2.5 µL (10 µM) of the primers (For-ATGTTTATGATCACAGCGGT and Rev-ATTGGGTTGCACCACTTGTC). PCR was performed on a ProFlex™ PCR Thermal Cycler (Applied Biosystems^®®^) following the next thermal conditions: denaturation at 95 °C for 4 min, 35 cycles of 95 °C for 30 s, at 50.8 °C for 30 s, and 72 °C for 1 min, and a final extension at 72 °C for 5 min. Water was employed as a negative control and cDNA prepared from a commercial vaccine (Nobivac Puppy—MSD Animal Health, Summit, NJ, USA) was used as a positive control.

For rt-PCR positive samples, the Fsp-coding region and a portion the H gene and were screened using Maxima Hot Start Green PCR Master Mixture. The H gene was amplified using the primers forward (CDVff1-TCGAAATCCTATGTGAGATCACT) and reverse (CDVHS2-ATGCTGGAGATGGTTTAATTCAATCG) for a 2099 bp fragment [17]. The Fsp-coding region was amplified using the primers F5/R5 (For-TGTTACCCGCTCATGGAGAT and rev—CCAAGTACTGGTGACTGGGTCT) flanking the F gene [18]. For all PCR, 6 µL cDNA was added to the reaction mixture, which consisted of 25 µL Master Mixture (2X), 15 µL nuclease-free water, 2 µL (10 µM) of forward and 2 µL (10 µM) of reverse primers. Thermal conditions included a denaturation at 95 °C for 4 min and 35 cycles at 95 °C for 30 s, annealing for 30 s, and 72 °C for 2 min, and a final extension at 72 °C for 10 min. The annealing temperature for H PCR was 48.2 °C, and 50.8 °C for the F PCR.

For PCR confirmation, 5 µL amplicon were run on a 1.5% agarose gel (AGAROSE I™, Amresco, OH, USA) electrophoresis and visualized by EZ-VISION™ (Amresco, OH, USA) under UV light on the GelDoc TM XR + System (Bio-Rad, Hercules, CA, USA). Amplicon size was estimated using the GeneRuler™ 100 bp Plus DNA Ladder (Thermo Scientific®®, Glen Burnie, MD, USA). For sanger sequencing, PCR amplicons were submitted to Macrogen Inc. (Seoul, Korea) for purification and sequencing. As previously reported [7], sequencing primers were used in a ABI3711™ automatic sequencer (Macrogen™, Seoul, Korea).

### 2.6. Phylogenetic Analysis

Sequence data were edited and assembled by using SeqMan software (DNAStar Lasergene™ V15.0 software package, Madison, WI, USA). Nucleotide BLAST was used for studying similarity between Colombian Fox CDV sequences and CDV sequences accessible on the Genbank database. An H gene sequence was obtained only for the Fox-3 sample (CT/Zorro3/CO/2021 Genbank ON458033). The Fsp-coding region were obtained for the Fox-2 and Fox-3 samples (CT/Zorro2/CO/2021 Genbank ON458031—CT/Zorro3/CO/2021 Genbank ON458032). Phylogenetic analyses were developed with a minimum of two sequences for each CDV lineage from different geographical areas using MEGA™ 7. Phylogenetic associations based on the nucleotide alignment of H gene and Fsp sequences were inferred using neighbor-joining and Maximum likelihood approaches implemented in MEGA™ 7. The best-fit model for nucleotide substitution was inferred by MEGA™ 7 as T92 + G for the H gene and HKY + G for the Fsp-coding region. Maximum likelihood assessment was accomplished by using those models. As an outgroup, we used the America-1 lineage to root the phylogenetic trees. Consensus trees were built in FigTree software version 1.4.

## 3. Results

### 3.1. Animals

On the clinical evaluation in the VHT, the animals were lethargic, paraplegic, and febrile (40–41C), with poor body condition (2/5), dehydration, mucopurulent conjunctivitis, and diarrhea in one individual. All other physical examination findings were considered normal. CBC and serum biochemistry were within normal limits in all animals. Due to the clinical scenario presented in these three foxes, and the previous diagnosis of CDV in other wild animals in this region of Colombia, a rapid CDV test was carried out, resulting positive for this virus in the three individuals. Due to worsening of the condition and low probability of re-introduction to the wild, all three animals were humanely euthanized.

### 3.2. Molecular Confirmation and Phylogenetic Analysis

In order to confirm this diagnosis with a more sensitive test, rt-PCR tests were performed. All three animals were confirmed positive by “Universal” morbillivirus PCR.

From the samples, we only amplified and sequenced the full-length H gene from Fox-3. However. To confirm that all animals were infected with the same virus, a 405 bp of the Fsp-coding region was assessed and amplified in samples belonging to the Fox-2 and Fox-3. We were unable to analyze any Fox-1 sequences due to low Q16 and Q20 Q scores. The Fox-3 H sequence displayed high identity with Colombian strains from North/South America-4 Lineage (93–99.1% nt; 97.1–99.3% aa); the same pattern was found for the Fsp-coding region (data no shown). Interestingly, the highest identity was to the KJ747371 and KJ747372 isolated from a fox and a dog from the United States. Phylogenetic relationships based on the nucleotide alignment of complete H gene and the Fsp sequences inferred by distance (neighbor-joining—data not shown) and character (maximum likelihood) approaches (Figure 2) produced trees with similar topology and showed the classical CDV phylogeographical distribution pattern.

Remarkably, we demonstrated that Colombian *Cerdocyon thous* CDV sequences cluster in the so-called South America/North America-4 lineage in both the H gene (Figure 2) and Fsp trees (Appendix A). In this clade, we found Colombian sequences clustering in the same clade as Ecuadorian and Peruvian strains with two North America-4 lineage sequences. Peruvian sequences (MT350712 to MT350717) grouped also in the South America/North America-4 lineage with les that 4% divergence at the a.a. level (2.3–3.1% to Colombian sequences). No other CDV South America/North America-4 lineage sequences from wild fauna were found in GenBank belonging from South America.

On the analysis of the mutational hotspots at the wildlife/domestic dog interface in the H protein, the current Fox-3 sequence displayed an aspartic acid at position 530, similar to the Peruvian and North American sequences. In contrast, the previous canine H Colombian CDV sequences isolated from naturally infected dogs presented a serine in this position. On the analysis of the position 549 of the H protein, we found that Fox-3 CDV has the same Tyrosine profile to the other Colombian reported sequences.

## 4. Discussion

Canine distemper has become one of the most important and devasting diseases for wildlife worldwide [10]. Although multiple outbreaks of CDV have been anecdotally recorded in the *Cerdocyon thous* population from Colombia, this is the first molecular and phylogenetic confirmation of CDV on wildlife from Colombia, and the first confirmation of the South America/North America-4 lineage infecting wildlife in the South American subcontinent [5,8,19]. The results of this study can contribute to a better understanding of the dynamics of CDV in populations of wild and domestic animals in the American continent and this information would serve to implement prevention and control measures for this important multi-host pathogen.

The clinical signs due to CDV infection in wild species are similar to those described in domestic dogs. The clinical severity of CDV depends on the environmental condition, host age, specie and immune status, and virulence of the strain. It is considered that between 50 and 70% of CDV infections in canids are subclinical, characterized by non-specific symptoms or mild self-limiting respiratory disease [20]. In all species, the respiratory, gastrointestinal, integumentary and CNS systems are most common affected [21]. In the case of the three foxes in this study, the clinical signs were similar to those described in the literature, except for the integumentary system, which was not affected.

In domestic dogs infected with CDV, abnormal CBC findings may include absolute lymphopenia due to depletion. Regenerative anemia and thrombocytopenia have been described in experimentally infected neonates; however, this has not been consistently recognized in adult or naturally infected dogs [4]. Other authors reported that the results of a CBC may be normal [22]. Hyperglobulinemia and hypoalbuminemia may be present [4]. Intriguingly, the results of the blood tests and biochemistry were normal in the three foxes. After CDV infection, the virus spreads rapidly to the oropharyngeal lymphoid organs, where it then enters the first of two viremic phases. During the first phase, there is generalized immune suppression due to the infection of all lymphoid tissues, fever, anorexia, and respiratory symptoms. During the second viremic phase, frequently associated with high fever, the virus is systemically disseminated and results in infection of parenchymal and tissue cells throughout the body reaching the gastrointestinal tract, central nervous system, and skin among other tissues [23,24]. Although no histopathology confirmation of CDV infection was possible on this cases, active plasma viremia and clinical signs such as high fever and neurological impairment allow to confirm the disease in the foxes.

Based on population studies in the Bolivian Chaco, it has been proposed that the Crab-eating foxes may be less susceptible to infection with CDV, or highly susceptible to fatal disease and unlikely to survive and produce antibodies [25]. A population-based survey of CDV in Colombian wild canid population would help to get a better understanding of the transmission dynamics of CDV and other important canid pathogens.

The lineage South America/North America-4 is one of the few known intercontinental lineages for the CDV [8]. This lineage first appeared in 2011 and was detected in dogs from multiple states in the southeast region of the United States and then detected from 2011 to 2013, including wildlife submissions [18]. Importantly, neutralizing serologic testing showed significant differences in neutralizing antibody titers between this strain and the strain commonly used in vaccines [26]. Subsequently, in 2014 this lineage was described for the first time in Ecuadorian samples from dogs [19], confirmed in 2019 in Colombian dog samples [8] and more recently in 2021 in Perú [27]. The transmission bridge between North and South America has not been established; however, it has been proposed the establishment of wild animal reservoirs that can move the CDV through central America not necessarily involving domestic dog transmission cycles [12]. A recent phylodynamic analysis has stated that North American CDV strains migrated to Ecuador in the early 1960s with posterior spread to the neighboring countries such as Peru and Colombia [28].

As a multi-host pathogen, the CDV must adapt mutations that allow the viral hemagglutinin to interact with the new host cell receptor. Either generalist or specialist strains adapt different mutational profiles at the SLAM-binding sites of the haemagglutinin gene that help to fit better to the new host [14]. Substitutions at residues 530 and 549 are associated with CDV isolated from novel host species [29]. Substitutions at position 530 (G/E to R/D/N) and 549 (Y to H) from the hemagglutinin has been seen to the spread of domestic dog adapted CDV strains to other carnivores and wild canid hosts are more frequently infected by strains with 549Y [30]. The profile 530D/549Y of the Fox-3 strain confirmed the interspecies profile of the current strain is the same as the North American-4 strains isolated from wildlife. This mutation pattern should be assumed a baseline for future outbreak studies in wild and domestic canids to understand and propose pathways of virus transmission and interspecies jump [14]. Since the functional role of substitutions at position 549 in CDV host tropism been proved in vitro [31,32] and the reported hypothesis that the dominant cross-species pathways of transmission during an outbreak can be studied through the analysis of the residue at site 549; thus, if the reservoir is a canid species, then tyrosine might be expected; if the reservoir is not a canid, then histidine might be expected [33]. We could propose that CDV infection in wild/domestic interface in Colombia are currently driven by wild canid reservoirs to unvaccinated domestic dog populations.

The *Cerdocyon thous* (Linnaeus, 1766) is a medium-sized nocturnal carnivore (3–8 kg) belonging to the family *Canidae* that is widely distributed over the Neotropical Region of the Americas (Colombia, Venezuela, Guyana, Surinam, French Guiana, Paraguay, Uruguay, northern Argentina and the greater part of Brazil) [34]. It is a generalist mammalian species associated with anthropized environments which is tolerant to modified habitats and due to its omnivorous diet and adaptability to modified environments, it can frequently be in close contact to humans [35,36]. The proximity to urban areas can not only modify the diet habits of the *Cerdocyon thous;* contact with domestic dogs increased the risk of transmission of infectious agents that are not frequent in wildlife. It can act as a bridge between domestic and wild environments and allow the transmission of pathogens between domestic animals and wildlife [37].

CDV may also be circulating independently in wildlife [14]. Epizootics in wildlife can result in a spillover to domestic dogs, and CDV strains that circulate between domestic dogs and raccoons has been reported [38]. In this context, it has been reported that the North/South America-4 lineage is highly prevalent in wildlife from eastern Tennessee, United States, reporting the presence of the virus in animals with (86%) and without (55%) clinical signs, with the majority (77%) testing positive for the South America/North America-4 lineage [39]. The confirmation of this same CDV lineage in Colombian wildlife enhances the importance to investigate the role of CDV transmission in wild fauna and the possible spillover and spillback from/to domestic dogs. It has been well-established that periodic outbreaks in raccoons can lead to a spillover/spillback into domestic dogs and spillover to other wildlife species [38]. Artic foxes have been the confirmed viral source in a CDV outbreak in sled dogs in Greenland [40]. In Germany, free-ranging carnivores (especially red foxes) may serve as a source of CDV infection to domestic dogs and vice versa [41]. The autonomous circulation of CDV in wild animals remarks the probable position as a “CDV reservoir” that those species can have, with the likelihood of spreading those CDV strains to unvaccinated domestic animals and the existence of successive epidemic outbreaks, as has been recently reported for the co-circulation of two CDV lineages in domestic dogs and wildlife from Central Italy [42] and for the North-Eastern regions in Italy experiencing severe and widespread recurring outbreaks of CDV affecting the wild carnivore and domestic dog populations [43].

Multiple reports of CDV infection in *Cerdocyon thous* has been established in South America; however, this is the first report of the infection in Colombian wild animals. A serosurvey of CDV in *Cerdocyon thous* from Emas National Park, Brazil, between November 2000 and May 2008 showed a 12.1% rate of infection [44]. In the southern region of Brazil a serosurvey found 25% of sampled animal positive for CDV [45] and phylogenetic analysis from an animal showing neurological signs has confirmed the presence of the Europe/South America-1 lineage in the Brazilian wild *Cerdocyon thous* population [46]. Additionally, in samples from animals from El Palmar National Park, Argentina, either dead or exhibiting clinical neurologic signs, the analysis showed the presence of a CDV strain had a high percentage of identity compared to CDV strains affecting dogs [47]. Moreover, the recent report of the virome present in *Cerdocyon thous* run over by cars from southern Brazil and Uruguay has demonstrated the common presence of CDV in animals and the phylogenetic analysis confirmed that CDV in those animals showed 99.3 to 99.6% identity with CDV strains detected in domestic dogs in Brazil [48]. These data highlight the strong role of *Cerdocyon thous* and other sympatric species as a possible reservoir for CDV in the Americas.

Although the *Cerdocyon thous* is a least-concern species [34], its biological behavior can turn it into a risk bridge for the transmission of CDV to endangered species. CDV is a well-known potential threat to vulnerable populations. CDV infection increased the 50-year extinction probability of Amur tigers (*Panthera tigris altaica*) up to 55.8% compared to a control population, depending on risk scenario [49]. Recent studies on the domestic dog and wildlife populations on the Russian Far East showed that wildlife species are more important than dogs, both for the maintaining the infection and as a CDV source of infection for Siberian tigers [50,51]. These results allow us to hypothesize that CDV infected foxes can be a source for CDV infection in endangered fauna (carnivorous and non-carnivorous) from Colombia.

## 5. Conclusions

Beyond domestic dogs, CDV is still a risk for the conservation of multiple wild animal populations. The infection of CDV in *Cerdocyon thous* highlights the importance of the understanding of the wild/domestic interface in the transmission of CDV and the important role that sympatric populations have in the interspecies transmission of viral diseases.

## Figures and Tables

**Figure 1 viruses-14-01947-f001:**
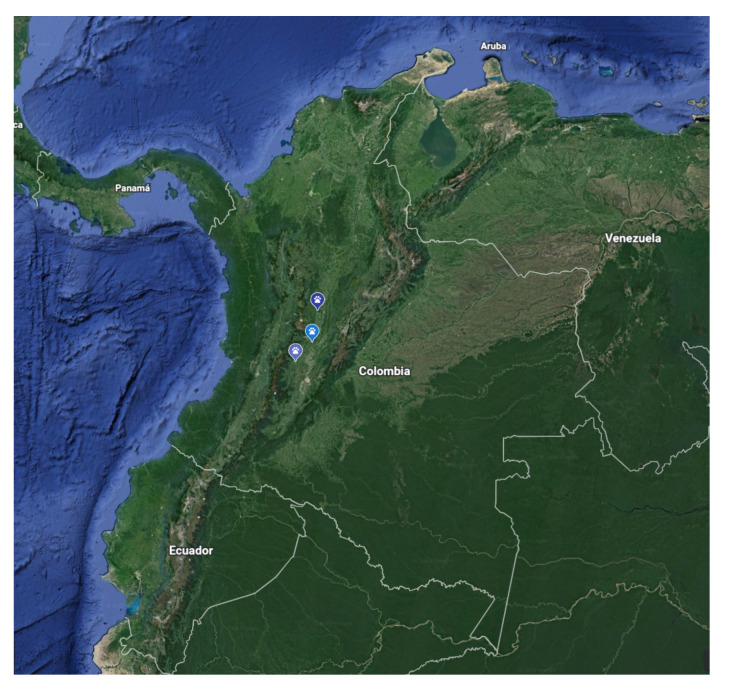
Map of location of CDV positive Crab-eating foxes in Colombia (locations designated by marks).

**Figure 2 viruses-14-01947-f002:**
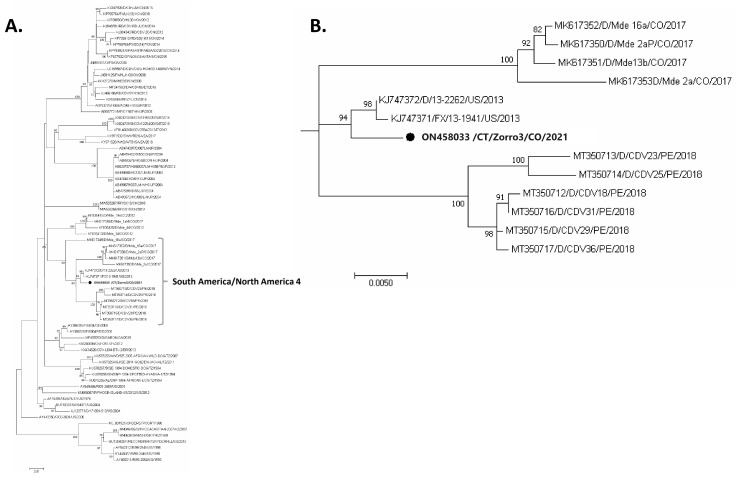
Phylogenetic Maximum likelihood tree for CDV H gene using 1000 bootstrap. Sequence labels include accession numbers, infected species, strain, country, and year. Bootstrap support is indicated at nodes. (**A**) full tree (**B**) South America/North America-4 lineage subtree. Abbreviations for animal species: AL: African lion (*Panthera leo*), B: badger (*Meles meles*), CT: *Cerdocyon thous*, D: dog (*Canis lupus familiaris*), F: ferret (*Mustela putorius furo*), FX: fox *(Vulpes urocyon*), GJ: golden jackal (*Canis aureus*), GP: giant panda (*Ailuropoda melanoleuca*), J: javelina (*Tayassu pecari*), LP: lesser panda (*Ailurus fulgens*), M: mink (*Neovison vison*), MP: Martens (*Martes pennanti*), R: raccoon (*Procyon lotor*), RD: raccoon dog (*Nyctereutes procyonoides*), S: seal (*Phoca vitulina*), SH: spotted hyena (*Crocuta crocuta*). List of countries: AR: Argentina, AT: Austria, BR: Brazil, CN: China, CO: Colombia, DE: Germany, DK: Denmark, HU: Hungary, IT: Italy, JP: Japan, KR: South Korea, KZ: Kazakhstan, MX: Mexico, PE: Peru, SE: Sweden, TZ: Tanzania, US: United States, UY: Uruguay, ZA: South Africa. Black dot denotes Colombian *Cerdocyon thous* CDV sequence.

## Data Availability

Not applicable.

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
