# Peer review of "The First Report and Phylogenetic Analysis of Canine Distemper Virus in Cerdocyon thous from Colombia"

_viruses, 2022, doi:10.3390/v14091947_

Round 1

Reviewer 1 Report

The authors describe the detection of CDV in 3 Cerdocyon thous from Columbia. The full CDV H gene from 1 animal was sequenced. The CDV fsp region from this same animal and one other was sequenced, and phylogenetic analyses for both the H gene and the fsp region were performed. This is the first report of the South-America/North-America-4 lineage in wildlife in Columbia. However, the number of animals is quite small, so the impact of this lineage in this species (or other susceptible species in the region) is difficult to determine.

The major weaknesses of this work is the lack of pathology/histopathology results. CDV can infect wildlife and not produce clinical signs. One could argue that detection of CDV in the blood does not prove that CDV was the cause of the clinical signs in these animals. This needs to be addressed in the discussion.

There are many grammatical/syntax errors, some of which are noted. Please have a fluent English speaker review the manuscript.

Simple Summary:

Lines 20-21- We confirm for the first time the presence of the South America/North America-4 lineage of CDV in wildlife in Columbia. (You mention that CDV has been previously diagnosed, so I think suggesting this is the first confirmation of CDV in Columbian wildlife is incorrect.)

Line 23-control of such an important…

Abstract:

lines 24-27- run-on sentence. Suggest breaking this into 2 sentences.

Lines 27-28- …wildlife, to date molecular characterization of the circulating lineages has not been performed.

Line 30- same as simple summary- not the first time to detect CDV in Columbian wildlife, unless previous detections were not confirmed by CDV testing

Introduction:

Line 49: To date,…

Line 50- multiple families and species of animals including the endangered…

Line 51- multi-host

Lines 52-53- grammar issue- Do you mean CDV can be endemic in wildlife species, even in the absence of cases in domestic dogs?

Line 54- CDV outbreaks in lions were shown to be…

Lines 56-57- meta-reservoirs is still unclear for most…

Line 58- of CDV lineage…

Line 61- wildlife, to date…

Lines 63-65- Our aim was to evaluate the clinical features associated with canine distemper in crab-eating foxes (Cerdocyon thous) in Colombia and to compare molecular and phylogenetic relationships between the virus sequences from these foxes with previously reported CDV sequences from domestic dogs…

M/M

Lines 68-72- three sick adult crab-eating foxes were found and reported to the Regional Ambiental Authority… Each fox was found in a different municipality of the Tolima Department… (Figure 1).

Line 72-73- remove this clinical exam info- found in results section

Line 74- attending veterinarian

Lines 74-77- remove this info about the serum samples. Redundant- found in the next section.

Figure 1. Map of location of CDV positive crab-eating foxes in Columbia (locations designated by marks).

Lines 82-105- All the details are unnecessary and these sections can be condensed. Blood samples were collected for a routine CBC, performed with an automated hematology analyzer (source), and a chemistry panel (creatinine, AST, ALT, and ALP), which was performed with an automated biochemical analyzer (source). Serum (plasma?) was also used for CDV testing.

Line 109- based on the description of the sample used for CDV testing, this should have been plasma, not serum. Was plasma used and is the test approved for this sample type?

Line 133- Water was used as a negative control and cDNA prepared from a commercial vaccine (which one, provide source) was used as a positive control.

Line 136- For RT-PCR CDV positive samples, the … and a portion of the H gene were amplified using the.…

Lines 150-151- Amplicon size was estimated using…

Results:

Lines 173-175- Move the name of the VTH to the methods section and change wording- Upon clinical presentation, the animals were lethargic, paraplegic, and febrile (40-41C), with poor body condition (2/5), dehydration, ….

Line 177- within normal limits

Line 179- move to a separate sections, to match with the section in M/M or combine sections in the M/M.

Line 187-Provide results for Fox 1- unable to sequence?

Line 189- pattern

Line 190- data not shown

Line 191- fox and a dog from the United States

Line 193-195- inferred by a distance (neighbor-joining- data not shown) and a character (ML) approach (Figure 2) showed the classical…

Lines 209-211- The Colombian Cerdocyon thous CDV sequences clustered with South American/North American-4 lineage sequences for both the H gene (Figure 2) and the fsp region (Supplemtary Figure 1).

Lines 215-216- First report- Move to discussion- not results

Lines 217-222- correct errors in spelling and grammar

Discussion:

Lots of spelling and grammatical errors in the discussion. Please correct.

Lines 245-247- no pathology/histopath reported to confirm CDV was the cause of the clinical signs; there are subclinical infections with CDV.

Lines 308-322- consider removing paragraph- data from Brazil does not add anything to the discussion.

Author Response

RESPONSE TO REVIEWERS COMMENTS (Manuscript ID: viruses-1885952)

The first report and phylogenetic analysis of Canine distemper virus in Cerdocyon thous from Colombia by Diego Fernando Echeverry-Bonilla, Edwin Fernando Buriticá-Gaviria, Delio Orjuela-Acosta, Danny Jaír Chinchilla-Cardenas and Julian Ruiz-Saenz

Dear Editor,

Please find below our point-by-point responses to the comments regarding our Manuscript ID: viruses-1885952, entitled “The first report and phylogenetic analysis of Canine distemper virus in Cerdocyon thous from Colombia”. The changes are highlighted in Yellow in the file.

We would like to thank the Reviewers for their helpful suggestions, for critical analysis of the manuscript, and for providing new discussion topics.

REVIEWER 1

The authors describe the detection of CDV in 3 Cerdocyon thous from Columbia. The full CDV H gene from 1 animal was sequenced. The CDV fsp region from this same animal and one other was sequenced, and phylogenetic analyses for both the H gene and the fsp region were performed. This is the first report of the South-America/North-America-4 lineage in wildlife in Columbia. However, the number of animals is quite small, so the impact of this lineage in this species (or other susceptible species in the region) is difficult to determine.

R/. We agree to the reviewer. The number of animals is limited by the difficulty of wildlife sampling in Colombia. However We do believe that the phylogenetic result is strong enough by using two different set of Genes for sequencing in order to confirm the presence of the South-America/North-America-4 lineage in Cerdocyon thous from Colombia.

The major weakness of this work is the lack of pathology/histopathology results. CDV can infect wildlife and not produce clinical signs. One could argue that detection of CDV in the blood does not prove that CDV was the cause of the clinical signs in these animals. This needs to be addressed in the discussion.

R/. We agree to the reviewer. Although no histopathology confirmation of CDV infection was possible on this cases, active plasma viremia and clinical signs such as high fever and neurological impairment allow to confirm the disease in the foxes.

Short Paragraph was added to support our data (Lines 246-262)

There are many grammatical/syntax errors, some of which are noted. Please have a fluent English speaker review the manuscript.

R/. We apologize for the unintentional typos. The full manuscript was double checked by Native English speaker

Simple Summary:

Lines 20-21- We confirm for the first time the presence of the South America/North America-4 lineage of CDV in wildlife in Columbia. (You mention that CDV has been previously diagnosed, so I think suggesting this is the first confirmation of CDV in Columbian wildlife is incorrect.)

R/. We agree to the reviewer. The sentences were modified.

Line 23-control of such an important…

R/. The sentence was modified

Abstract:

lines 24-27- run-on sentence. Suggest breaking this into 2 sentences.

R/. The sentence was modified

Lines 27-28- …wildlife, to date molecular characterization of the circulating lineages has not been performed.

R/. The sentence was modified

Line 30- same as simple summary- not the first time to detect CDV in Columbian wildlife, unless previous detections were not confirmed by CDV testing

R/. The sentence was modified

Introduction:

Line 49: To date,…

R/. The sentence was modified in whole document.

Line 50- multiple families and species of animals including the endangered…

R/. The sentence was modified

Line 51- multi-host

R/. The sentence was modified

Lines 52-53- grammar issue- Do you mean CDV can be endemic in wildlife species, even in the absence of cases in domestic dogs?

R/. Yes. It has been confirmed the circulation of CDV in wildlife species in complete independence of domestic dog cases. This has been highlighted previously in other papers and has been related to the existence of specialist vs. generalist CDV strains (doi.org/10.3390/v11070582). The sentence was modified to clarify this item (Lines 51-53).

Line 54- CDV outbreaks in lions were shown to be…

R/. The sentence was modified

Lines 56-57- meta-reservoirs is still unclear for most…

R/. The sentence was modified

Line 58- of CDV lineage…

R/. The sentence was modified

Line 61- wildlife, to date…

R/. The sentence was modified

Lines 63-65- Our aim was to evaluate the clinical features associated with canine distemper in crab-eating foxes (Cerdocyon thous) in Colombia and to compare molecular and phylogenetic relationships between the virus sequences from these foxes with previously reported CDV sequences from domestic dogs…

R/. The sentence was modified according to the reviewer recommendation.

M/M

Lines 68-72- three sick adult crab-eating foxes were found and reported to the Regional Ambiental Authority… Each fox was found in a different municipality of the Tolima Department… (Figure 1).

R/. We agree to the reviewer. The sentences were modified.

Line 72-73- remove this clinical exam info- found in results section

R/. The sentence was modified

Line 74- attending veterinarian

R/. The sentence was modified

Lines 74-77- remove this info about the serum samples. Redundant- found in the next section.

R/. The sentence was modified

Figure 1. Map of location of CDV positive crab-eating foxes in Columbia (locations designated by marks).

R/. The sentence was modified

Lines 82-105- All the details are unnecessary and these sections can be condensed. Blood samples were collected for a routine CBC, performed with an automated hematology analyzer (source), and a chemistry panel (creatinine, AST, ALT, and ALP), which was performed with an automated biochemical analyzer (source). Serum (plasma?) was also used for CDV testing.

R/. We agree to the reviewer. The paragraphs were Condensed (Lines 79-100)

Line 109- based on the description of the sample used for CDV testing, this should have been plasma, not serum. Was plasma used and is the test approved for this sample type?

R/. The sentence was modified.  The sample used was Plama and the Kit has been validated for this sample (see: https://www.bionote.co.kr/en/product/rapid/view.html?idx=60)

Line 133- Water was used as a negative control and cDNA prepared from a commercial vaccine (which one, provide source) was used as a positive control.

R/. The sentence was modified and Vaccine information added.

Line 136- For RT-PCR CDV positive samples, the … and a portion of the H gene were amplified using the.…

R/. The sentence was modified

Lines 150-151- Amplicon size was estimated using…

R/. The sentence was modified

Results:

Lines 173-175- Move the name of the VTH to the methods section and change wording- Upon clinical presentation, the animals were lethargic, paraplegic, and febrile (40-41C), with poor body condition (2/5), dehydration, ….

R/. We agree to the reviewer. The paragraph was modified according to the recommendation.

Line 177- within normal limits

R/. The sentence was modified

Line 179- move to a separate sections, to match with the section in M/M or combine sections in the M/M.

R/. We agree to the reviewer. The paragraph was modified according to the recommendation.

Line 187-Provide results for Fox 1- unable to sequence?

R/. We were unable to analyze any FOX-1 sequences due to low Q16 and Q20 Q Scores. Information was added.

Line 189- pattern

R/. The sentence was modified

Line 190- data not shown

R/. The sentence was modified

Line 191- fox and a dog from the United States

R/. The sentence was modified

Line 193-195- inferred by a distance (neighbor-joining- data not shown) and a character (ML) approach (Figure 2) showed the classical…

R/. The sentence was modified

Lines 209-211- The Colombian Cerdocyon thous CDV sequences clustered with South American/North American-4 lineage sequences for both the H gene (Figure 2) and the fsp region (Supplemtary Figure 1).

R/. The sentence was modified

Lines 215-216- First report- Move to discussion- not results

R/. The sentence was modified to

Lines 217-222- correct errors in spelling and grammar

R/. We Agree to the reviewer. The paragraph was corrected

Discussion:

Lots of spelling and grammatical errors in the discussion. Please correct.

R/. We apologize for the unintentional typos. The full manuscript was double checked by Native English speaker review

Lines 245-247- no pathology/histopath reported to confirm CDV was the cause of the clinical signs; there are subclinical infections with CDV.

R/. We agree to the reviewer. Although no histopathology confirmation of CDV infection was possible on this cases, active plasma viremia and clinical signs such as high fever and neurological impairment allow to confirm the disease in the foxes. Short Paragraph was added to support our data (Lines 246-262)

Lines 308-322- consider removing paragraph- data from Brazil does not add anything to the discussion.

R/. We did not remove because conflict with another reviewer. However, the paragraph was edited to add more information to the distribution of the CDV in Crab eating foxes populations in South America.

Reviewer 2 Report

Dear authors,

The manuscript compared the clinical features, molecular characteristics, and phylogenetic relationships between Canine distemper virus found in clinical samples from sick animals and previously reported CDV sequences from domestic dogs in Colombia and the Americas.

 The paper is clearly written, experiments were properly designed, and the correct analysis was performed with their data.

 These are my questions, possible amendments, and recommendations to which I would ask you to refer.

 -        Line 120: Have RNA treated with DNase before using it as a template in cDNA synthesis? the treatment with DNase is very important to eliminate any genomic DNA; please clarify.

-        Line 120: what did you mean by (1-3) in 13 μl (1–3, 1 μg) total RNA

-        Line 129: what is the primer reference? is it synthesized during this study? what is the primer reference? is it synthesized during this study? please clarify all primers references used throughout the manuscript

-        Line 189: The same pattern was found…. instead of ; The same patter was found….. correct it

-        Line 197: improve the quality of figure 2 (A and B)

Author Response

RESPONSE TO REVIEWERS COMMENTS (Manuscript ID: viruses-1885952)

The first report and phylogenetic analysis of Canine distemper virus in Cerdocyon thous from Colombia by Diego Fernando Echeverry-Bonilla, Edwin Fernando Buriticá-Gaviria, Delio Orjuela-Acosta, Danny Jaír Chinchilla-Cardenas and Julian Ruiz-Saenz

Dear Editor,

Please find below our point-by-point responses to the comments regarding our Manuscript ID: viruses-1885952, entitled “The first report and phylogenetic analysis of Canine distemper virus in Cerdocyon thous from Colombia”. The changes are highlighted in Yellow in the file.

We would like to thank the Reviewers for their helpful suggestions, for critical analysis of the manuscript, and for providing new discussion topics.

REVIEWER 2

The manuscript compared the clinical features, molecular characteristics, and phylogenetic relationships between Canine distemper virus found in clinical samples from sick animals and previously reported CDV sequences from domestic dogs in Colombia and the Americas.

 The paper is clearly written, experiments were properly designed, and the correct analysis was performed with their data.

 These are my questions, possible amendments, and recommendations to which I would ask you to refer.

 -        Line 120: Have RNA treated with DNase before using it as a template in cDNA synthesis? the treatment with DNase is very important to eliminate any genomic DNA; please clarify.

R/. We agree to the reviewer. DNAse treatment is very useful. Unfortunately, we did not used. We used high quality RNA extracted by the Qiagen viral extraction kit and we also use the RiboLock RNase Inhibitor included to protects RNA templates from degradation. That procedure helps us to have high quality RNA templates.

-        Line 120: what did you mean by (1-3) in 13 μl (1–3, 1 μg) total RNA

R/. We apologize for this unintentional typo. The document was edited. 13 µl (1-3 µg) total RNA.

Total Volumen was 13 13 μl. Total RNA concentration was between 1 and 3 µg.

-        Line 129: what is the primer reference? is it synthesized during this study? what is the primer reference? is it synthesized during this study? please clarify all primers references used throughout the manuscript

R/We agree to the reviewer. The primer references and sequences were added

-        Line 189: The same pattern was found…. instead of ; The same patter was found….. correct it

R/. The sentence was modified and double check the entire document

-        Line 197: improve the quality of figure 2 (A and B)

R/ We agree to the Reviewer. The figure quality was enhanced to obtain a better presentation (800x800dpi)

Round 2

Reviewer 1 Report

Thank you for the edits. I have no further suggestions.